# Probing interlayer shear thermal deformation in atomically-thin van der Waals layered materials

Le Zhang[1,3], Han Wang[1,3], Xinrong Zong[2], Yongheng Zhou[1], Taihong Wang[1], Lin Wang [2✉] & Xiaolong Chen [1✉]

Atomically-thin van der Waals layered materials, with both high in-plane stiffness and bending flexibility, offer a unique platform for thermomechanical engineering. However, the lack of effective characterization techniques hinders the development of this research topic. Here, we develop a direct experimental method and effective theoretical model to study the mechanical, thermal, and interlayer properties of van der Waals materials. This is accomplished by using a carefully designed $WSe_2$-based heterostructure, where monolayer $WSe_2$ serves as an in-situ strain meter. Combining experimental results and theoretical modelling, we are able to resolve the shear deformation and interlayer shear thermal deformation of each individual layer quantitatively in van der Waals materials. Our approach also provides important interlayer coupling information as well as key thermal parameters. The model can be applied to van der Waals materials with different layer numbers and various boundary conditions for both thermally-induced and mechanically-induced deformations.

[1] Department of Electrical and Electronic Engineering, Southern University of Science and Technology, 1088 Xueyuan Avenue, 518055 Shenzhen, P.R. China. [2] Key Laboratory of Flexible Electronics (KLOFE) & Institute of Advanced Materials (IAM), Nanjing Tech University (Nanjing Tech), 30 South Puzhu Road, 211816 Nanjing, P.R. China. [3] These authors contributed equally: Le Zhang, Han Wang. ✉email: iamlwang@njtech.edu.cn; chenxl@sustech.edu.cn

Triggered by the growing need of developing next-generation semiconductor devices, mechanical engineering has been moved forward from traditional semiconductors to van der Waals (vdW) materials due to their unique layered structures[1,2]. Through lattice deforming, the electronic structure of vdW materials can be tuned significantly, giving rise to intriguing physical phenomena and applications, such as shear-strain-generated pseudo magnetic fields[3], one-dimensional moiré potentials[4], confined states in soliton networks[5], and actively variable-spectrum optoelectronics[6]. Mechanical approaches have been widely used to introduce compressive and tensile strain (lattice deformation) in vdW materials, including substrate engineering with nanopillars[7–9], generating nanobubbles in vdW materials[10–13], bending flexible substrates[14–16], and utilizing the thermal expansion coefficient (TEC) mismatch between vdW materials and substrates[17–19]. Although much progress has been achieved in the mechanical engineering of vdW materials, investigations on their thermomechanical properties are scarce. Besides, understanding of the micro-mechanism of interlayer deformation when reacting to temperature variation lies at the heart of thermal engineering of vdW materials.

Since vdW materials are always supported by substrates, their thermomechanical properties are considered based on a whole vdW-materials/substrate system. vdW materials and substrates generally possess distinct TEC, leading to distinct intrinsic thermal deformation when temperature changes. Figure 1a shows the schematic diagram of thermal deformation in an $N$-layer vdW-material/$SiO_2$ system from $T_0$ to $T_1$. Considering the strong clamping effect between $SiO_2$ and vdW materials[18], the deformation of the bottom layer ($n = 1$) is nearly equal to that of $SiO_2$. Yet the top layer ($n = N$) is almost free from the clamping effect and exhibits intrinsic thermal deformation of vdW materials when $N$ is large enough. In this case, the relaxation from layer to layer through interlayer interaction results in the in-plane lattice deformation difference between adjacent layers. For clarity, we define the shear thermal deformation (STD) $\tau$ and interlayer shear thermal deformation (ISTD) $\Delta\tau$ in Fig. 1a. Here, $\tau$ is the in-plane lattice deformation induced by temperature variation from $T_0$ (high temperature) to $T_1$ (low temperature) and $\Delta\tau$ is the in-plane lattice deformation difference between adjacent layers. However, owing to the lack of proper characterization technique, precise measurements of STD and ISTD layer by layer in vdW materials have not been reported yet.

In this work, we choose phosphorene and hexagonal boron nitride (hBN) as the representative experimental subjects due to their exceptional thermal and mechanical properties[18,20–22]. Through monitoring the temperature-dependent photoluminescence (PL) spectra of delicately designed $WSe_2$-based vdW heterostructures, where monolayer $WSe_2$ serves as an in-situ "strain meter", the mechanical behaviors of vdW materials are

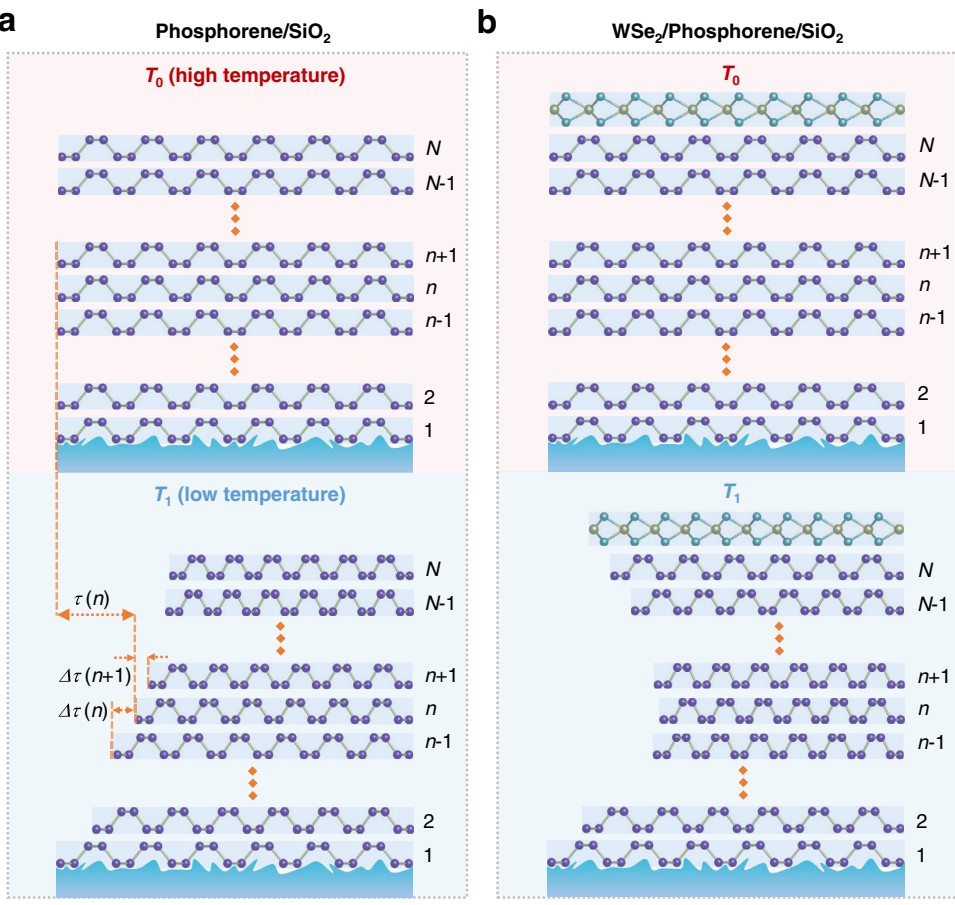

$\tau(n)$: shear thermal deformation (STD) of the $n$th layer
$\Delta\tau(n)$ : interlayer shear thermal deformation (ISTD)

**Fig. 1 Shear thermal deformation (STD) and interlayer shear thermal deformation (ISTD) in van der Waals materials and heterostructures. a** ISTD model of an $N$-layer phosphorene/$SiO_2$ system when temperature decreases from $T_0$ to $T_1$. **b** ISTD model of a $WSe_2$/$N$-layer phosphorene/$SiO_2$ system when temperature decreases from $T_0$ to $T_1$. Here, $N$ is the total layer number of phosphorene. $n$ is the $n$-th ($1 \leq n \leq N$) phosphorene layer counting from bottom. $\tau(n)$ is the STD of the $n$-th phosphorene and $\Delta\tau(n)$ is the ISTD between the $n$-th and ($n-1$)-th phosphorene from $T_0$ to $T_1$.

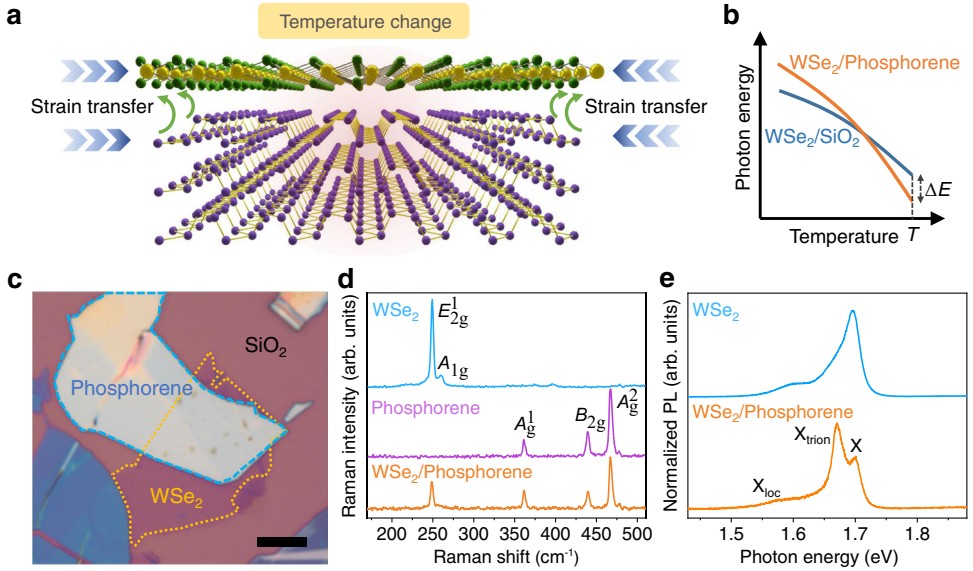

**Fig. 2 WSe$_2$/phosphorene heterostructure for STD and ISTD investigations. a** Monolayer WSe$_2$ serves as a strain meter to probe the thermal deformation of the top phosphorene layer. Here, the blue arrows indicate that the in-plane lattice contraction when temperature decreases. The green arrows describe the strain transfer from phosphorene to WSe$_2$ through interlayer interactions. **b** Schematic evolution trends of photon energy in WSe$_2$/phosphorene/SiO$_2$ and WSe$_2$/SiO$_2$ systems as a function of temperature. Photon energy difference ($\Delta E$) between WSe$_2$/phosphorene/SiO$_2$ and WSe$_2$/SiO$_2$ at temperature $T$ can directly reflect the thermal deformation of WSe$_2$. **c** Optical image of a WSe$_2$/phosphorene heterostructure. The scale bar is 5 μm. **d** Raman spectra of isolated monolayer WSe$_2$, isolated phosphorene, and WSe$_2$/phosphorene heterostructure at room temperature. **e** Photoluminescence (PL) spectra of isolated monolayer WSe$_2$ and WSe$_2$/phosphorene heterostructure at 180 K. Here, X$_{loc}$, X$_{trion}$ and X denotes the localized, charged and neutral exciton of monolayer WSe$_2$, respectively.

reflected conveniently. Taking account of interlayer interactions at both homo- and hetero-interfaces and the Young's modulus and TEC of phosphorene and WSe$_2$, we establish an effective ISTD model which allows us to access the layer-dependent STD and ISTD in phosphorene. The schematic diagram in Fig. 1b illustrates the thermal deformation reaction of phosphorene and WSe$_2$ layers in the WSe$_2$/phosphorene/SiO$_2$ system from $T_0$ to $T_1$. Through fitting the experimental results, we can extract the interlayer coupling coefficients at phosphorene/phosphorene homo-interface and WSe$_2$/phosphorene hetero-interface. Besides, key thermal parameters of vdW materials, such as TEC, are extracted from the model.

## Results

**Design of vdW heterostructures for ISTD studies**. We choose monolayer WSe$_2$ and phosphorene as building blocks of vdW heterostructures for STD and ISTD investigations for three reasons. First, phosphorene has been predicted to exhibit a large TEC of between $6.3 \times 10^{-6}$ and $53 \times 10^{-6}$ K$^{-1}$ at room temperature[23–25], which stands out from the family of vdW materials and is at least one order of magnitude larger than that of SiO$_2$ ($\sim 0.5 \times 10^{-6}$ K$^{-1}$)[26,27]. Such a large TEC is expected to cause significant thermal deformation and corresponding effect on the physical properties of phosphorene and its adjacent 2D materials. Second, monolayer WSe$_2$ is a flexible direct-bandgap semiconductor with very high luminescence efficiency[28]. Its strain-sensitive optical and electronic properties have been widely investigated[16,17,28–30]. Thus, we can use monolayer WSe$_2$ as a convenient sensing layer to monitor the thermal deformation of phosphorene as illustrated in Fig. 2a. Third, phosphorene and WSe$_2$ show strong coupling at their interface, which could enable efficient strain transfer from phosphorene to WSe$_2$ through interlayer interactions (Fig. 2a)[31].

To quantitatively investigate the STD and ISTD of phosphorene, we monitor the temperature-dependent PL photon energy of

WSe$_2$/phosphorene/SiO$_2$ and use that of WSe$_2$/SiO$_2$ as a reference system, since the TEC of SiO$_2$ can be neglected compared with that of phosphorene[23–26]. Even though WSe$_2$ in WSe$_2$/SiO$_2$ and WSe$_2$/phosphorene/SiO$_2$ experiences different dielectric environments which could affect the exciton binding energies of WSe$_2$, in this study we are focusing on $\Delta E' = \Delta E(T_1) - \Delta E(T_0)$, where $\Delta E(T)$ is the relative shift of photon energy in WSe$_2$/phosphorene/SiO$_2$ compared with that in WSe$_2$/SiO$_2$ at temperature $T$ (see Fig. 2b). The effect of dielectric environment plays a minor role in determining $\Delta E'$ as shown in Supplementary Fig. 1 and Supplementary Note 1. As illustrated in Fig. 2b, $\Delta E'$ directly reflects the STD of WSe$_2$ in the temperature range from $T_0$ to $T_1$. Considering the interlayer coupling effect at the WSe$_2$/phosphorene interface, information of STD and ISTD in phosphorene layers can be extracted.

**Characterizations of WSe$_2$/phosphorene heterostructures**. Figure 2c shows the optical image of a WSe$_2$/phosphorene heterostructure, assembled by the PDMS-assisted dry-transfer method[32]. To achieve a high-quality vdW interface, all exfoliation and transfer processes were performed in a N$_2$-filled glove box. The cross-sectional scanning transmission electron microscopy (STEM) image and elemental mapping demonstrate a clean and amorphous-phase-free WSe$_2$/phosphorene interface (see Supplementary Fig. 2 and Supplementary Note 2). The Raman spectra collected from isolated monolayer WSe$_2$, isolated phosphorene, and WSe$_2$/phosphorene heterostructure are displayed in Fig. 2d, respectively. The characteristic phonon vibration modes of the monolayer WSe$_2$ ($E_{2g}^1$)[33] and few-layer phosphorene ($A_g^1$, $B_{2g}$, and $A_g^2$)[34–36] are all observed in the heterostructure region. Figure 2e shows the PL spectra of the isolated WSe$_2$ and WSe$_2$/phosphorene heterostructure at 180 K, where layer numbers of the phosphorene are 50 ($\sim$27.5 nm) determined by atomic force microscope. Three pronounced photon emission peaks, located at 1.59, 1.67, and 1.70 eV, are observed. They can be attributed to

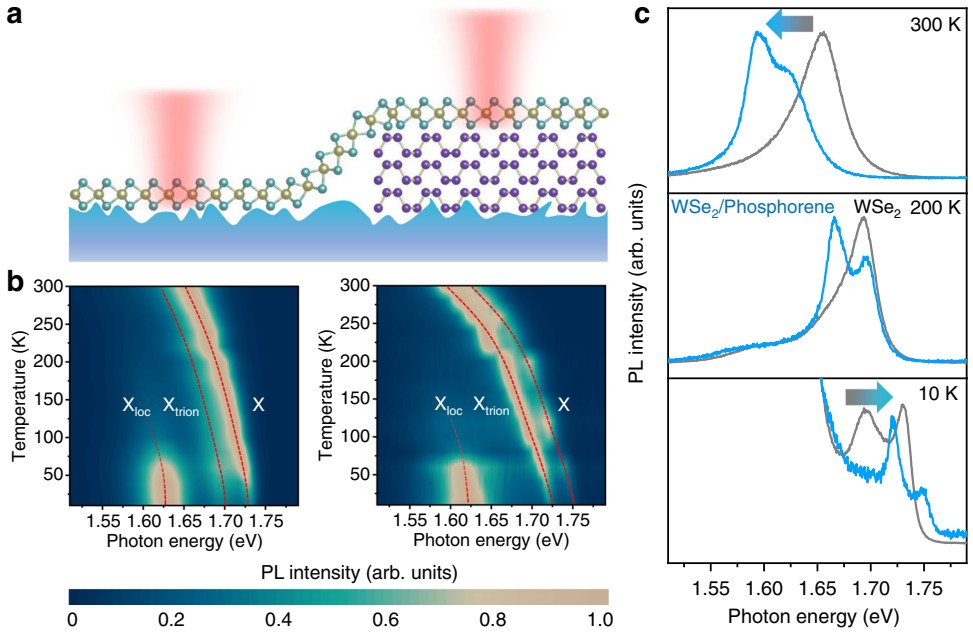

**Fig. 3 Temperature-dependent PL characterizations. a** Schematic diagram of PL characterizations on WSe$_2$/SiO$_2$ and WSe$_2$/phosphorene/SiO$_2$ regions. **b** PL of neutral exciton (X), charged exciton (X$_{trion}$) and localized exciton (X$_{loc}$) in isolated WSe$_2$ on SiO$_2$ substrate and WSe$_2$/phosphorene heterostructure from 10 to 300 K. The red dashed lines serve as guide lines. **c** The normalized PL spectra of isolated WSe$_2$ (gray lines) and WSe$_2$/phosphorene heterostructure (blue lines) at 10, 200, and 300 K. Here, the gray-blue arrows indicate that the photon energy of X and X$_{trion}$ in WSe$_2$/phosphorene/SiO$_2$ heterostructure is blue-shifted/red-shifted relative to those in WSe$_2$/SiO$_2$.

localized exciton (X$_{loc}$), charged exciton (X$_{trion}$), and neutral exciton (X)[37,38], respectively. The X$_{trion}$ emission is more pronounced in WSe$_2$/phosphorene heterostructure which can be attributed to the charge transfer at vdW interface[39].

Then we explore the temperature-dependent properties of PL in the isolated WSe$_2$ on SiO$_2$ and WSe$_2$/phosphorene heterostructure (see the schematic diagram in Fig. 3a). As shown in Fig. 3b, both X and X$_{trion}$ show energy shift when temperature changes. Moreover, X and X$_{trion}$ of WSe$_2$/phosphorene heterostructure go through a greater shift than those of the isolated WSe$_2$ on SiO$_2$, indicating that additional strain is induced to the WSe$_2$ on phosphorene when temperature changes. Figure 3c further compares the normalized PL of isolated WSe$_2$ and WSe$_2$/phosphorene heterostructure at three representative temperatures. The photon energy of X and X$_{trion}$ in WSe$_2$/phosphorene heterostructure is larger than that in WSe$_2$ at 10 K, the same at 200 K, and smaller at 300 K. With temperature changing, the top phosphorene layer performs greater in-plane lattice deformation than SiO$_2$. Therefore, an additional strain is transferred into WSe$_2$ on phosphorene through vdW interlayer interactions, leading to the greater shift of photon energy relative to that of WSe$_2$ on SiO$_2$. According to previous research results, tensile/compressive strain in monolayer WSe$_2$ will reduce/enlarge its optical bandgap[16,17,28–30], showing excellent agreement with our observations.

**ISTD model and experimental measurement.** To confirm our expectations, an ISTD model considering interlayer coupling effect is established to quantitatively determine the STD and ISTD in vdW materials. When temperature changes, the in-plane interlayer interaction is generated between adjacent layers due to lattice deformation mismatch (i.e., the ISTD). This produces additional in-plane stress in individual layer. Here, the in-plane stress is linked to ISTD through a proportionality factor, which is defined as the interlayer coupling coefficient ($c$) between adjacent layers. $c_p$ and $c_h$ denote the interlayer coupling coefficient at

phosphorene/phosphorene homo-interface and WSe$_2$/phosphorene hetero-interface, respectively.

Taking $T_0$ as the initial state, the strain in each layer is 0 (see Fig. 1). As shown in Fig. 1a, the STD of the $n$-th phosphorene ($1 \leq n \leq N$, counting from the bottom layer) from $T_0$ to $T_1$ is noted as $\tau(n)$. When $2 \leq n \leq N-1$, the $n$-th phosphorene interacts with the $(n-1)$-th as well as with the $(n+1)$-th phosphorene layers. Therefore, $\tau(n)$ satisfies the following equation at $T_1$ [Eq. 1]:

$$c_p(\Delta\tau(n+1) - \Delta\tau(n)) = \gamma_p(\tau(n) - \tau_p) \qquad 2 \leq n \leq N-1 \qquad (1)$$

Here, $\Delta\tau(n)$ is the ISTD between the $n$-th and $(n-1)$-th phosphorene (Fig. 1a). $\gamma_p$ is the Young's modulus of phosphorene, which is around 60 GPa according to previous works[40]. $\tau_p$ is the thermally-induced intrinsic deformation of phosphorene, which is a constant depending on the TEC of phosphorene.

The cases of phosphorene/SiO$_2$ (Fig. 1a) and WSe$_2$/phosphorene/SiO$_2$ (Fig. 1b) share the same boundary condition for $n = 1$ (at phosphorene/SiO$_2$ interface). When $n = 1$, considering the strong clamping effect between the bottom phosphorene and SiO$_2$ substrate[18] as well as the tiny TEC of SiO$_2$[26,27], an approximation can be made that the STD of the bottom phosphorene is negligible, that is, $\tau(1) = 0$. When $n = N$, the mechanical behaviors of the top phosphorene are totally different in phosphorene/SiO$_2$ and WSe$_2$/phosphorene/SiO$_2$ systems. In WSe$_2$/phosphorene/SiO$_2$ system, the interlayer interaction at phosphorene/WSe$_2$ interface and the mechanical and thermal properties of WSe$_2$ have direct impact on the STD of phosphorene layers. Please refer to Supplementary Note 3 for the details, where the interlayer coupling coefficient between phosphorene and WSe$_2$ ($c_h$), the Young's modulus of WSe$_2$ ($\gamma_{WSe_2}$) and thermally-induced intrinsic deformation of WSe$_2$ ($\tau_{WSe_2}$) are introduced. Here, $c_h$ is calculated as $2.72 \times 10^{11}$ Pa, $\gamma_{WSe_2}$ is assigned to be 120 GPa according to previous reports[41–45], and $\tau_{WSe_2}$ is extracted to be $-0.17\%$ at 10 K (see Supplementary Note 4).

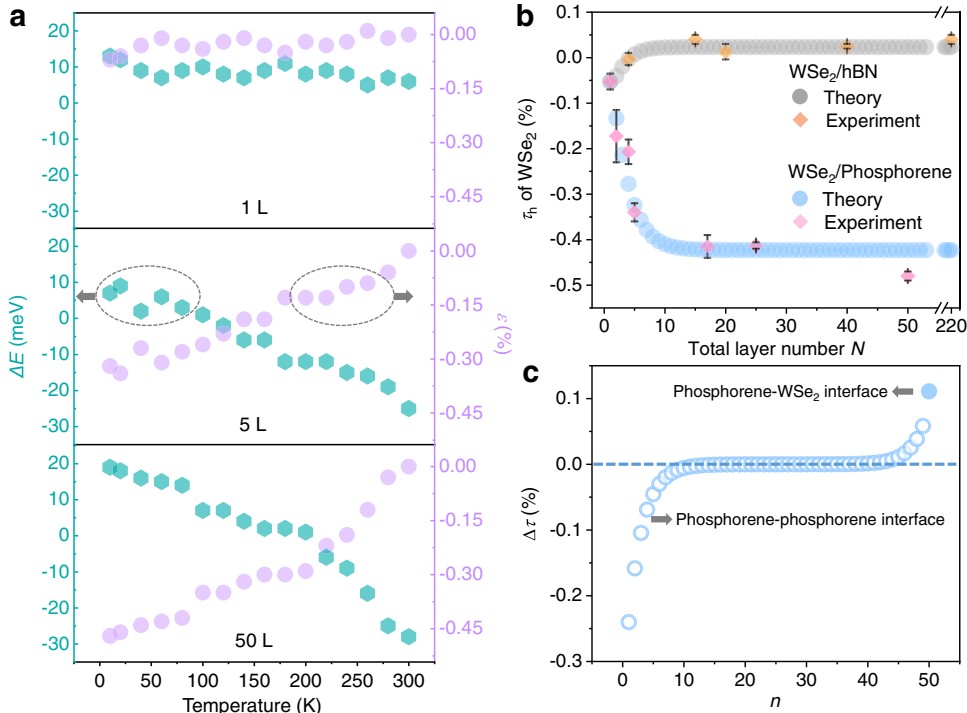

**Fig. 4 Extracting STD and ISTD from experiment results and theoretical modelling. a** Cyan symbols show the temperature-dependent photon energy difference ($\Delta E$) between WSe$_2$/phosphorene and isolated WSe$_2$ at phosphorene layer numbers of 1, 5, and 50, respectively. Violet symbols shows the experimentally measured strain of WSe$_2$ as a function of temperature. **b** From 300 to 10 K, experimentally measured and theoretically fitted STD of WSe$_2$ ($\tau_h$) in WSe$_2$/phosphorene and WSe$_2$/hBN systems with different total layer number $N$. Here, for each sample, PL signals are collected from at least three spots near the center of heterostructures. The average value (pink/orange symbols) and standard deviation (error bar, black solid line) of $\tau_h$ ($N$) are therefore obtained from the measured $\Delta E$. **c** Theoretically calculated layer-dependent ISTD ($\Delta\tau$) in the WSe$_2$/50-layer phosphorene/SiO$_2$ system. Here, the empty circles denote $\Delta\tau$ at phosphorene-phosphorene interface while the filled circle denotes $\Delta\tau$ at phosphorene-WSe$_2$ interface.

To confirm the validity of our theory, we experimentally characterize WSe$_2$/phosphorene heterostructures with various phosphorene layer number $N$ from 1 to 50. Figure 4a shows measured $\Delta E$ as a function of temperature at three representative layer numbers of 1, 5, and 50 (cyan symbols), respectively. Then, taking advantage of $\eta = -100$ meV/% (see Supplementary Fig. 3, Supplementary Table 1 and Supplementary Note 5), where $\eta$ is the coefficient of strain-induced energy shift in monolayer WSe$_2$, the additional in-plane strain of WSe$_2$ in the heterostructure $\varepsilon = \Delta E/\eta$ can be extracted (violet symbols in Fig. 4a). The fitting results at other $\eta$ values are further shown and compared in Supplementary Table 2.

Taking 300 K as the initial temperature $T_0$ and 10 K as the final temperature $T_1$, the STD of WSe$_2$ on $N$-layer phosphorene, $\tau_h(N) = (\Delta E(T_1) - \Delta E(T_0))/\eta$, is shown in Fig. 4b. The measured STD (pink symbols) agrees well with the theoretical fitting results (blue symbols), with fitting parameters $c_p = 3.41 \times 10^{11}$ Pa and $\tau_p = -0.71\%$. The error analysis of the theoretical fitting results is provided in Supplementary Fig. 4 and Supplementary Note 6. Here, the larger interlayer coupling coefficient $c_p$ at phosphorene/phosphorene homo-interfaces than $c_h$ at WSe$_2$/phosphorene hetero-interfaces indicates the stronger coupling at phosphorene/phosphorene homo-interfaces. When $N > 15$, $\tau_h$ is almost independent of layer number and reaches the minimum value of $-0.42\%$. This phenomenon is in accordance with our expectations since the substrate clamping effect is weaker for top phosphorene layers when $N$ is larger. Utilizing the fitting results above, we are able to calculate the $n$-dependent STD and ISTD quantitatively in WSe$_2$/phosphorene/SiO$_2$ systems with various total layer number $N$. Figure 4c shows the $n$-dependent ISTD ($\Delta\tau$) in a WSe$_2$/50-layer phosphorene

heterostructure. With $n$ increasing, there appears a crossover point where the negative $\Delta\tau$ turns into positive. The slope is steeper near the phosphorene/SiO$_2$ and WSe$_2$/phosphorene hetero-interfaces due to strong mismatch of in-plane strain. Besides, we can access the layer-dependent in-plane force in phosphorene and WSe$_2$ layers as shown in Supplementary Fig. 5. The in-plane force is large near phosphorene/SiO$_2$ and WSe$_2$/phosphorene interfaces (at order of 0.1 N m$^{-1}$) while vanishes in the central region.

Utilizing the fitting results of $c_p = 3.41 \times 10^{11}$ Pa and $\tau_p = -0.71\%$, STD and ISTD of phosphorene in phosphorene/SiO$_2$ system can be also calculated (see Supplementary Note 3). Here, we compare the calculation results of phosphorene/SiO$_2$ and WSe$_2$/phosphorene/SiO$_2$ systems with different $N$ in Fig. 5. Figure 5a and d are the schematic diagrams of the two cases at $N = 5$. In phosphorene/SiO$_2$ system, STD of phosphorene ($\tau$) decreases monotonously and nonlinearly with $n$ (see Fig. 5b). When $N > n > 15$, $\tau$ is almost independent of $n$ and reaches the minimum value of $-0.71\%$, which approaches the intrinsic thermal deformation of phosphorene at 10 K. On the other hand, in WSe$_2$/phosphorene/SiO$_2$ system, STD of phosphorene initially decreases and then increases from bottom to top, revealing the non-uniform deformation near the WSe$_2$/phosphorene interface (Fig. 5e). When $N$ is large enough, $\tau$ reaches the minimum value $-0.71\%$ in the middle region ($17 < n < 28$ for $N = 50$) and increases to $-0.53\%$ at $n = N$. In addition, $\Delta\tau$ of phosphorene increases monotonously with $n$ and approaches zero when $N > n > 15$ in phosphorene/SiO$_2$ system (Fig. 5c), while in WSe$_2$/phosphorene/SiO$_2$ system, $\Delta\tau$ turns into positive values near the WSe$_2$/phosphorene interface due to the small TEC and large Young's modulus of WSe$_2$ (Fig. 5f). The distinctions between the two cases directly reflect the interlayer coupling effect between WSe$_2$ and phosphorene.

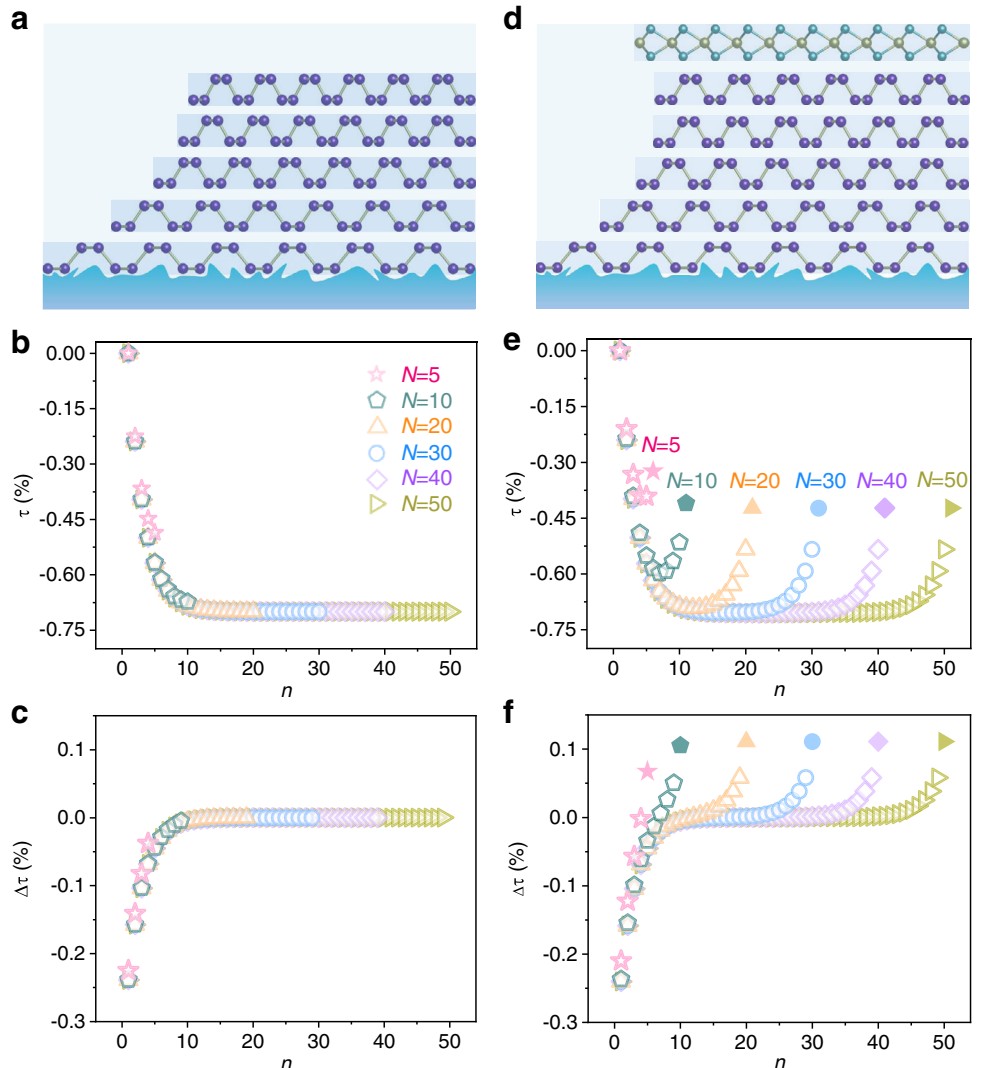

**Fig. 5 Resolving STD and ISTD of individual layer in phosphorene/SiO₂ and WSe₂/phosphorene/SiO₂ systems. a** Schematic diagram of 5-layer phosphorene/SiO₂ under thermomechanical deformation at low temperature. **b, c** Theoretically calculated layer-dependent $\tau$ (**b**) and $\Delta\tau$ (**c**) in phosphorene/SiO₂ at representative layer numbers $N = 5$, 10, 20, 30, 40 and 50. **d** Schematic diagram of WSe₂/5-layer phosphorene/SiO₂ at at low temperature. **e, f** Theoretically calculated layer-dependent $\tau$ (**e**) and $\Delta\tau$ (**f**) in WSe₂/phosphorene/SiO₂ at representative layer numbers $N = 5$, 10, 20, 30, 40, and 50. Here, the empty symbols denote values in phosphorene while the filled symbols denote values in WSe₂.

**TEC of phosphorene**. To our best knowledge, experimental studies on the TEC ($\alpha$) of atomically-thin vdW layered materials are scarce, all of which are based on the high-temperature X-ray diffraction technique[23,25,46], temperature-dependent Raman and electron energy-loss spectroscopy[21,27]. In the following section, we are going to show that the additional strain measured in WSe₂/phosphorene heterostructures can be utilized to achieve this goal.

Utilizing the temperature-dependent additional strain of WSe₂ in the WSe₂/50-layer phosphorene (Fig. 4a), the intrinsic thermal strain of phosphorene as a function of temperature can be extracted through the ISTD model (violet symbols in Fig. 6a). A local band average approach under Debye approximation has declared that the temperature-dependent TEC is proportional to the specific heat ($C_v$) and can be expressed as[47] [Eq. 2]:

$$\alpha(T) = A(T/\theta_D)^3 \int_0^{\theta_D/T} \frac{x^4 e^x}{(e^x - 1)^2} dx \qquad (2)$$

where $A$ is a constant and $\theta_D$ is the Debye temperature. Then the thermal strain of the phosphorene can be obtained by integrating TEC [Eq. 3]:

$$\varepsilon(T) = \int \alpha(T) dT \qquad (3)$$

According to previous reports, $\theta_D = 600$ K is used to conduct the fitting[48]. As shown in Fig. 6a, the fitted results (blue solid line) match well with the experimental results (violet symbols) with parameter $A = 1.64 \times 10^{-4}$ K$^{-1}$, indicating the validity of our ISTD model. Figure 6b shows the extracted TEC of phosphorene from our theoretical model. The TEC is very small at low temperature and increases to $4.52 \times 10^{-5}$ K$^{-1}$ at 300 K. The obtained TEC ~ $4.52 \times 10^{-5}$ K$^{-1}$ at room temperature agrees quite well with previously reported values at high temperature (>300 K)[23–25].

**STD and TEC of hBN**. Distinct from phosphorene, hBN possesses a negative TEC instead, whose absolute value is an order of magnitude smaller than that of phosphorene according to

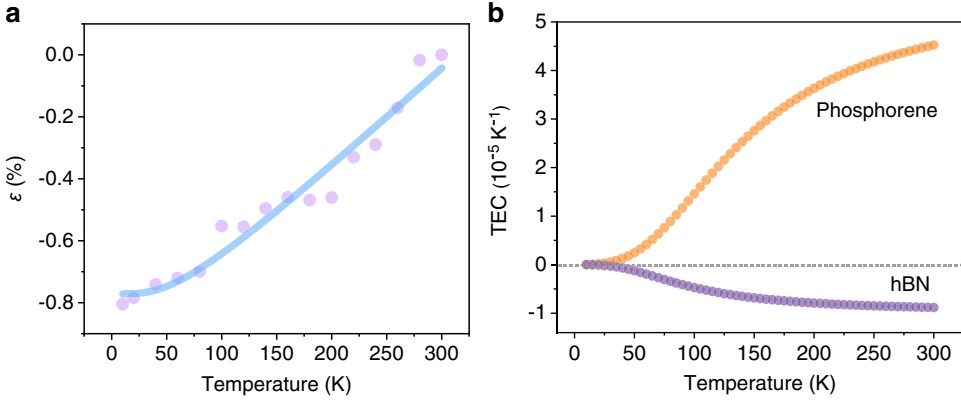

**Fig. 6 Thermal parameters of phosphorene and hBN. a** Experimentally measured (violet symbols) and theoretically fitted (blue solid line) intrinsic thermal strain of phosphorene as a function of temperature. **b** Extracted temperature-dependent thermal expansion coefficient (TEC) of phosphorene and hBN.

previous studies[21,46]. Therefore, to confirm the validity of our ISTD model, we repeat experiments based on WSe$_2$/hBN heterostructures. The layer-dependent STD of hBN $\tau$ $(n)$ is extracted and plotted in Fig. 4b, with fitting parameters $c_{hBN} = 5.03 \times 10^{11}$ Pa and $\tau_{hBN} = 0.17\%$. STD of WSe$_2$ in WSe$_2$/hBN and WSe$_2$/phosphorene shows distinct trend with $N$. This phenomenon is in accordance with the small and negative TEC of hBN. Then, the WSe$_2$/220-layer hBN heterostructure is adopted to extract TEC of hBN utilizing $\theta_D = 410$ K[46]. The temperature-dependent TEC of hBN is plotted in Fig. 6b (purple symbols). The extracted TEC is $-8.83 \times 10^{-6}$ K$^{-1}$ at room temperature, showing good agreement with reported values[21,46]. Therefore, the investigation of hBN further confirms the validity of our ISTD model and the effectiveness of the technique for sensing thermal properties of vdW layered materials.

## Discussion

At last, we discuss the novelty of this work and the distinction from previous works[13,49,50]. First, we provide a smart strategy, the WSe$_2$-based heterostructure, to investigate the mechanical, thermal, and interlayer coupling properties of vdW materials. Second, the ISTD model can quantitatively resolve the interlayer deformation in individual layers of vdW materials and heterostructures with various layer numbers. Third, the model can provide important interlayer coupling information, such as the interlayer coupling coefficients and in-plane force at phosphorene/phosphorene homo-interface and WSe$_2$/phosphorene hetero-interface. Fourth, the intrinsic-thermal-deformation-induced strain is more stable, reversible, and controllable compared with mechanical bending/stretching, allowing us to provide a clearer physical picture of interlayer shear deformation and extract key thermal parameters accurately. Last, the model can be applied and extended to various deformation situations (both thermally-induced and mechanically-induced deformations) with various boundary conditions, which can be easily modified and used by other researchers. Hence, we believe the smart experimental methodology, the ISTD model, the clear physical picture and the interlayer coupling information provided in this work will inspire thermomechanical engineering in vdW materials and be beneficial to the scientific community of 2D materials.

## Methods

**Sample preparation**. Monolayer and few-layer phosphorene/hBN flakes were mechanically exfoliated from bulk crystals (purchased from HQ graphene) onto 285 nm SiO$_2$/Si substrate through the standard scotch tape method. To avoid material degradation, the exfoliation process was carried out in a N$_2$-filled glovebox (MIKROUNA-Universal Series) with O$_2$ and H$_2$O concentrations smaller than 0.01 ppm. The monolayer WSe$_2$ was exfoliated onto PDMS substrate. Then the WSe$_2$/phosphorene and WSe$_2$/hBN heterostructures were assembled using the

PDMS-assisted dry-transfer method in the glovebox. We intentionally left half WSe$_2$ flake on SiO$_2$ as a reference to extract the thermal deformation of phosphorene and hBN. To improve the interlayer coupling between WSe$_2$ and phosphorene/hBN, the samples are annealed at 200 °C for 10 min inside the glovebox.

**Optical characterizations**. To carry out the temperature-dependent PL measurements, the samples were loaded in a He-flow closed-cycle cryostat (Advanced Research System) with a high vacuum of ~2 × 10$^{-6}$ Torr. A 532 nm laser was used as the excitation source and focused on the sample by a 50× objective lens (NA = 0.5). The laser power was kept at a low value of 25 μW to avoid laser-induced damage and heating effect. The PL signals were dispersed by an Andor SR-500i-D2 spectrometer with a 150 g/mm grating and detected using an Andor iVac 316 CCD. As for the Raman measurements, which were conducted at room temperature, a 600 g/mm grating was used.

## Data availability

Relevant data supporting the key findings of this study are available within the paper and the Supplementary Information file. All raw data generated during the current study are available from the corresponding authors upon request.

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

## Acknowledgements

The authors acknowledge the STEM support of Xin Zhou from National University of Singapore and Chao Zhu from Southeast University, and SUSTech Core Research Facilities. The work was financially supported by the open research fund of Songshan Lake Materials Laboratory (2021SLABFN02, X.C.), the National Natural Science Foundation of China (Grant No. 61904077, X.C; 92064010, L.W.; 61801210, L.W.; 91833302, L.W.), the National Key R&D Program of China (Grant No. 2020YFA0308900, L.W.), the funding for "Distinguished professors" and "High-level talents in six industries" of Jiangsu Province (Grant No. XYDXX-021, L.W.).

## Author contributions

X.C. and L.W. conceived and supervised the projects. L.Z., H.W. and X.Z. fabricated $WSe_2$/phosphorene and $WSe_2$/hBN heterostructure samples. L.Z. performed temperature-dependent photoluminescence characterizations with assistance of H.W., X.Z., and Y.Z. L.Z. and X.C. analyzed the experimental data. H.W. and X.C. did the theoretical modelling. H.W. performed Raman characterizations. X.C., L.Z., L.W., H.W. and T.W. drafted the paper. All authors discussed and commented the paper.

## Competing interests

The authors declare no competing interests.
