## [Peer Review File · Nature Communications]

Probing interlayer shear thermal deformation in atomically-thin van der Waals layered materialsREVIEWER COMMENTS

Reviewer #1 (Remarks to the Author):

This is potentially an interesting paper but is fundamentally flawed. The thermal expansion of thin van der Waals layered materials, consisting of layers of phosphorene of different thickness on a rigid SiO₂ substrate with a top monolayer of WSe₂, is investigated.

A major mistake is that it is assumed that the WSe₂ layer is very flexible and monitors the strain on the top of the phosphorene layers (p8). The Young's modulus of atomically thin WSe₂ is 149.1 GPa (ACS Nano 15 2021 2600) and is 58.6 GPa and 27.2 GPa in the zig-zag and armchair directions of few-layer phosphorene (ACS Nano 9 2015 11362). This means that the WSe₂ cannot be considered to be a flexible strain sensor with no effect upon the behaviour, as it is 3-4 times stiffer than the phosphorene. The WSe₂ will affect the strain in the phosphorene, particularly for thin layers, and render their analysis incorrect. In order to properly analyse the behaviour being reported, the authors will need to take into account the TEC of both the WSe₂ and the phosphorene as well as their respective Young's moduli (that are potentially temperature dependent). I think it will still be correct to assume that the substrate is infinitely stiff and has a low TEC.

Another issue that arises is the claim by the authors that the phenomenon ISTD they propose to investigate is new. Interlayer shear in 2D materials is well established during mechanical deformation, e.g. the reversible loss of Bernal stacking in few-layer graphene (ACS Nano 7 2013 7287). The only difference here is the effect of temperature. It is also odd that the authors claim the ISTD to be "rather giant" (p4) - a strange use of English.

Reviewer #2 (Remarks to the Author):

The manuscript by Zhang and co-workers reports on a temperature-dependent photo-luminescence study of 2-d materials hetero-structures. The authors quantify the relative changes of photon energies to determine the local lattice constant, thermal expansion and stresses. Overall, the paper is well-written, the results are carefully presented and the assumptions underlying the modeling appear solid. I recommend that the current manuscript be considered for publication once the authors address the following points:

- 1) The authors utilize their models to extract the thermal expansion coefficient, the Debye temperature and interlayer shear thermal deformation of the 2d vdW systems. Overall, it seems that they can match their results reasonably well to the previously published literature values. However, a careful analysis and description of error bars and measurement uncertainties are missing. Without such error discussions, it is difficult to assess how good the authors' models really are.
- 2) Have the authors confirmed, by using for example cross-sectional imaging in TEM, that the hetero-interfaces are clean? Is there no oxide (or other amorphous) interfacial layer that might affect the coupling between the substrate and the vdW layers?
- 3) The authors assume that there is nearly 100% coupling at the hetero-interfaces. Was that confirmed? Can that be confirmed? How would a less than ideal coupling affect the results?

Reviewer #3 (Remarks to the Author):

The manuscript by Le Zhang et al. discusses the interlayer shear strain in few-layer vdW materials. The authors developed an interesting idea to measure the strain of a few-layer 2D material by placing a monolayer WSe₂ on top and using its strain-sensitive optical properties. They utilize their method to measure the thermal expansion of few-layer black phosphorus and hBN.

I see a couple of issues with the manuscript which need to be addressed:

1. In Fig. 1a the authors claim that the relaxation of the stress induced by the different thermal expansion coefficient of substrate and 2D material (which they call shear thermal deformation) from layer to layer is a novel thermomechanical phenomenon. While the measurement of the relaxation from layer to layer is new, the general picture is trivial. The reason that this effect has been neglected (not ignored) is, that it is rarely of high relevance in 2D heterostructures (e.g. in typical encapsulated hBN/1L TMD/hBN samples) or not measurable.

2. While the authors claim that the strain is not fully transferred from layer to layer, they controversially assume "100 % strain transfer efficiency at the WSe₂/BP heterointerface" and "Thirdly, the attaching of WSe₂ has ignorable influence on the thermal deformation of phosphorene, in which case WSe₂ merely plays the role of strain sensor.", which definitely not true. Here, the authors need to discuss the Young's modulus of the BP and WSe₂ (which are pretty different) as well as the vdW coupling. In fact, on thin (1L, 2L,...) BP samples, the "probe" WSe₂ should have a significant impact on the expansion of the BP, so the TEC of the WSe₂ needs to be considered as well.

3. Furthermore, it is obvious, that the comparison between WSe₂ on SiO₂ and WSe₂ on BP fails already due to the different dielectric environments. Therefore, "Taking WSe₂ on SiO₂ as a reference system," is not a valid argument. However, the relative measurements on the differently thick BP should be ok.

4. A value of the strain gauge factor $\eta = 100 \text{ meV}/\%$ for WSe₂ is taken from the literature. It needs to be discussed if this value is true for the whole temperature range from 10 to 300 K.

5. A minor point: The definition: "Firstly, the strain of the bottom phosphorene is negligible, considering the strong clamping effect between phosphorene and SiO₂ substrates¹⁸ and tiny TEC of SiO₂ substrate³³." is a bit counter intuitive from my point of view. I would rather argue that the SiO₂ transfers a strong (compressive) stress/strain to BP at low temperatures while the top layer of a high enough bulk BP is fully relaxed (i.e. no strain), especially, since the sample is prepared at room temperature, which means that the sample should be fully relaxed at 300 K (in contrast to the schematic drawing in Fig. 3c).

6. Everything said above is also true for the hBN measurement.

In conclusion: The measurement of the strain transfer (or strain relaxation) from layer to layer is interesting and can have implications for other groups working on strain engineering of 2D heterostructures. In principle the measurements should also give access to the interlayer vdW coupling / stacking order energies.

However, I do not see a broader interest as the fundamental effect is rather trivial. Furthermore, I see several critical issues with the manuscript that require a major revision.

Point-by-point Response to Reviewers' Comments

We are grateful for the opportunity and guidance you gave us to revise our paper. We carefully considered and incorporated all of your valuable inputs in our revisions. As a result, we believe the paper has improved significantly. In this letter, we will first provide an overview of the major changes we made and then respond to your comments (reproduced in italics) following the order they appear in your reports.

Overviews of major changes

We have made the following important changes in the current version to address your main comments and suggestions.

First, the interlayer shear thermal deformation (ISTD) model is significantly improved through including the interlayer coupling coefficient at WSe₂/phosphorene (c_h) and phosphorene/phosphorene (c_p) interfaces, and the Young's modulus and thermal expansion coefficient (TEC) of both WSe₂ and phosphorene. Based on the new ISTD model, we not only obtain key mechanical and thermal parameters of phosphorene and hBN, but also extract important interlayer interaction information. Hence, the new results and new methodology make the paper more appealing to scientific community of nanomaterial and nanotechnology.

Second, instead of the continuous approximation used in previous version, we have used the discrete method to calculate the layer-dependent interlayer deformation in the current version. This allows us to obtain more accurate results. Besides, necessary additional experiments have been conducted to support our conclusions.

Third, we have clarified the novelty of our work with previous research works on mechanical deformations of vdW materials. 1) We provide a new strategy, the WSe₂-based vdW heterostructure, to investigate the interlayer coupling, mechanical and thermal properties of vdW materials. 2) Our model can quantitatively resolve the deformation value for each individual layer. 3) Our model can provide important interlayer coupling information, such as the interlayer coupling coefficients at phosphorene/phosphorene homo-interface and WSe₂/phosphorene hetero-interface. 4) Our model can be applied or extended to various boundary conditions for both thermal-induced and mechanical-induced deformation. 5) The thermal-induced strain used in our experiments is more stable, reversible, and controllable than mechanically approaches, allowing us to provide a clear physical picture of interlayer shear deformation and extract key thermal parameters accurately.

Finally, we have made every effort to fully address all of your other concerns and suggestions.

Point-by-point response to referee 1

We would like to thank you for your thoughtful comments and suggestions. We truly appreciate the time and efforts you invested in our paper. With your help, our paper has improved substantially. In this letter, we will respond to each of your comments following the order in which they appear in your report.

This is potentially an interesting paper but is fundamentally flawed. The thermal expansion of thin van der Waals layered materials, consisting of layers of phosphorene of different thickness on a rigid SiO₂ substrate with a top monolayer of WSe₂, is investigated.

1. A major mistake is that it is assumed that the WSe₂ layer is very flexible and monitors the strain on the top of the phosphorene layers (p8). The Young's modulus of atomically thin WSe₂ is 149.1 GPa (ACS Nano 15 2021 2600) and is 58.6 GPa and 27.2 GPa in the zig-zag and armchair directions of few-layer phosphorene (ACS Nano 9 2015 11362). This means that the WSe₂ cannot be considered to be a flexible strain sensor with no effect upon the behaviour, as it is 3-4 times stiffer than the phosphorene. The WSe₂ will affect the strain in the phosphorene, particularly for thin layers, and render their analysis incorrect. In order to properly analyse the behaviour being reported, the authors will need to take into account the TEC of both the WSe₂ and the phosphorene as well as their respective Young's moduli (that are potentially temperature dependent). I think it will still be correct to assume that the substrate is infinitely stiff and has a low TEC.

Response: Thank you for pointing out these important issues to us. We agree with you that the interlayer coupling between WSe₂ and phosphorene must be considered when analyzing the shear thermal deformation (STD) and interlayer shear thermal deformation (ISTD) of phosphorene.

In the revised manuscript, we have improved the ISTD model through taking account into the interlayer coupling coefficient at WSe₂/phosphorene (c_h) and phosphorene/phosphorene (c_p) interfaces, and the Young's modulus and thermal expansion coefficient (TEC) of WSe₂ and phosphorene (see **Figure R1** below).

$\tau(n)$: shear thermal deformation (STD) of the n th layer
 $\Delta\tau(n)$: interlayer shear thermal deformation (ISTD)

Figure R1. The previous (left panel) and new (right panel) interlayer shear thermal deformation (ISTD) models for $\text{WSe}_2/\text{phosphorene}/\text{SiO}_2$ heterostructures. The new model considers the interlayer interactions between WSe_2 and phosphorene.

Because WSe_2 is no longer treated as a flexible strain sensor, the boundary condition at $\text{WSe}_2/\text{phosphorene}$ interface has also been rewritten. Besides, we use accurate discrete method instead of previous continuous approximation method to conduct the calculations. The detailed simulation process is shown in **Supplementary Note 3 and Note 4**. Here, we compare the simulation results of STD and ISTD using previous and new ISTD models at several representative layer numbers $N = 5, 10, 20, 30, 40$ and 50 as shown in **Figure R2**. Comparing Figure R2a,b and Figure R2c,d, the new model clearly reveals the non-uniform deformation of phosphorene layers near the $\text{WSe}_2/\text{phosphorene}$ interface, which is the consequence of the interlayer coupling effect between WSe_2 and phosphorene layer and the large Young's modulus of WSe_2 . As a result, there appears a crossover point where the negative $\Delta\tau$ turns into positive as shown in Figure R2d.

Figure R2. Layer-dependent shear thermal deformation (τ) and interlayer shear thermal deformation ($\Delta\tau$) of atomically thin layer in WSe₂/phosphorene heterostructure using previous (a, b) and new theoretical models (c, d). Here, the hollow dots denote values in phosphorene while the solid dots denote values in WSe₂.

In addition to the STD, ISTD and TEC of phosphorene, the new model also provides information about the interlayer coupling coefficient (c_p , c_h) and the in-plane force induced in phosphorene and WSe₂. The simulation results give $c_p = 3.41 \times 10^{11}$ Pa and $c_h = 2.72 \times 10^{11}$ Pa. **Figure R3** shows the layer-dependent in-plane force in phosphorene and WSe₂ layers. The force is large near the hetero-interface while vanishes in the central region.

Figure R3. Layer-dependent in-plane force in phosphorene (hollow circle) and WSe₂

(solid circle) layers extracted from the new model.

In the revised manuscript, we have included Figure R1 and Figure R2 in the main text (see Figure 1 and Figure 5 in the main text). Figure R3 has been included in Supplementary Information (see Supplementary Figure5). All other changes and discussions of the new model have been highlighted in Page 7-13 in the main text and shown in Supplementary Note 3 and 4.

2. Another issue that arises is the claim by the authors that the phenomenon ISTD they propose to investigate is new. Interlayer shear in 2D materials is well established during mechanical deformation, e.g. the reversible loss of Bernal stacking in few-layer graphene (ACS Nano 7 2013 7287). The only difference here is the effect of temperature. It is also odd that the authors claim the ISTD to be "rather giant" (p4) - a strange use of English.

Response: Thank you for the valuable suggestions. We have avoided the use of “rather giant” and “new” in the revised manuscript. Besides, we have clarified the novelty of our work and compared it with previous works.

First, ACS Nano 7, 7287, 2013 work reported the observation of the change of Raman shape in a 3-layer graphene under mechanical stress¹. They explained the observation **qualitatively** using the interlayer shear effect. In our work, we provide an effective theoretical model, which can **quantitatively** resolve the STD and ISTD in individual layers in vdW layered materials (see **Figure R2** above). The model can be applied to vdW materials with various layer numbers N and various boundary conditions. The model can also provide important interlayer coupling information, such as c_p and c_h , which are not accessible in previous ACS Nano work.

Second, the intrinsic-thermal-expansion-induced strain used in our experiments is more stable, reversible and controllable than mechanical approaches, allowing us to provide a clearer physical picture of interlayer shear deformation and extract key thermal parameters accurately. Besides, the model can be applied and extended to various deformation situations (both thermal-induced and mechanical-induced deformation) with various boundary conditions, which can be easily modified and used by other researchers.

In addition, we provide a smart experimental strategy, using WSe₂-based heterostructures, to investigate key mechanical, thermal and interlayer properties of vdW materials. Hence, we believe the new experimental methodology, the new model, the clear physical picture and the interlayer coupling information provided in this work will be beneficial to the scientific community of 2D materials.

In the revised manuscript, we have included the work ACS Nano 7, 7287, 2013 in the reference list. Discussions on the novelty of the work and comparison with previous

works are shown in Page 13-14 in the main text.

Point-by-point response to referee 2

We would like to thank you for your thoughtful comments and suggestions. We truly appreciate the time and efforts you invested in our paper. With your help, our paper has improved substantially. In this letter, we will respond to each of your comments following the order in which they appear in your report.

The manuscript by Zhang and co-workers reports on a temperature-dependent photo-luminescence study of 2-d materials hetero-structures. The authors quantify the relative changes of photon energies to determine the local lattice constant, thermal expansion and stresses. Overall, the paper is well-written, the results are carefully presented and the assumptions underlying the modeling appear solid. I recommend that the current manuscript be considered for publication once the authors address the following points:

1. The authors utilize their models to extract the thermal expansion coefficient, the Debye temperature and interlayer shear thermal deformation of the 2d vdW systems. Overall, it seems that they can match their results reasonably well to the previously published literature values. However, a careful analysis and description of error bars and measurement uncertainties are missing. Without such error discussions, it is difficult to assess how good the authors' models really are.

Response: Thank you very much for the positive comments and valuable suggestions. In the revised manuscript, we have re-analyzed the error bars and re-plotted figures.

First, the main experimental errors originate from the measurement uncertainties of the relative shift of photon energy (ΔE) through photoluminescence (PL) measurement. We characterize WSe₂-based heterostructures with various phosphorene and hBN layer number N . For each sample, PL signals are collected from at least three spots near the center of heterostructures. Hence, the average value and standard deviation of ΔE can be obtained. Then the average value and standard deviation of thermal deformation (τ_h) of WSe₂ can be obtained (see **Figure R4** below), since τ_h is proportional to ΔE .

Figure R4. Experimental measured and theoretical shear thermal deformation (τ_h) of WSe₂ as a function of phosphorene/hBN layer number (N).

Second, a careful error analysis of theoretical model fitting has been provided in the revised manuscript. In the revised manuscript, we have improved the theoretical model through taking account into the intrinsic thermal deformation of phosphorene (τ_p), interlayer coupling coefficient at WSe₂/phosphorene (c_h) and phosphorene/phosphorene (c_p) interfaces. Here, c_h is directly calculated as 2.72×10^{11} Pa, while $c_p = 3.41 \times 10^{11}$ Pa and $\tau_p = -0.71\%$ are extracted through model fitting (using the least-square method). According to the standard deviation (0.0006) of the fitted τ_h , upper and lower bounds of c_p and τ_p can be determined and are listed in **Table R1**. **Figure R5** further shows the fitting results at upper and lower bounds of c_p and τ_p .

	Fitted value	Upper bound	Lower bound
c_p	3.41×10^{11} Pa	4.80×10^{11} Pa	2.39×10^{11} Pa
τ_p	-0.71 %	-0.65 %	-0.77 %

Table R1. Parameters of c_p and τ_p .

Figure R5. **a**, Theoretically fitted shear thermal deformation (τ_h) of WSe₂ as a function of N at upper and lower bounds of τ_p . **b**, Theoretically fitted τ_h of WSe₂ as a function of N at upper and lower bounds of c_p .

Third, in the manuscript, the biaxial strain gauge factor η is adopted as -100 meV/%, which stands between previous reported experimental (60 - 100 meV/%) and theoretical values (~ 130 meV/%). Hence, the uncertainty of η also affect the fitting results. To further study the influence of strain gauge factor on the fitting results, the strain gauge factor is set to be -80 , -100 and -120 meV/%. As shown in Table R2, both the interlayer coupling coefficients at WSe₂/phosphorene (c_h) and phosphorene/phosphorene (c_p) interfaces increase with $|\eta|$. Besides, c_h shows stronger η dependence than c_p . On the contrary, the intrinsic thermal deformation of phosphorene $|\tau_p|$ and TEC decrease with $|\eta|$.

η (meV/%)	c_h (Pa)	c_p (Pa)	τ_p	TEC at 300 K (K ⁻¹)
-80	1.94×10^{11}	3.22×10^{11}	-0.999 %	6.38×10^{-5}
-100	2.72×10^{11}	3.41×10^{11}	-0.705 %	4.52×10^{-5}
-120	3.50×10^{11}	3.63×10^{11}	-0.535 %	3.45×10^{-5}

Table R2. Fitting results when the biaxial strain gauge factor of WSe₂ is set to be -80, -100 and -120 meV/%, respectively.

In the revised manuscript, we have included Figure R4 in the main text (see Figure 4b) and Table R1, Table R2, and Figure R5 in the Supplementary Information (see Supplementary Table 3, Supplementary Table 1, and Supplementary Figure 4). The discussion of η is shown in Supplementary Note 5.

2. Have the authors confirmed, by using for example cross sectional imaging in TEM, that the hetero-interfaces are clean? Is there no oxide (or other amorphous) interfacial layer that might affect the coupling between the substrate and the vdW layers?

Response: Thank you for the valuable suggestions. We have fabricated a WSe₂/phosphorene heterostructure for cross-sectional scanning transmission electron microscopy (STEM) characterizations. Here, graphene serves as the protective layer to avoid possible damage of sample during shipping (the sample is mailed to National University of Singapore for STEM characterizations). Figure R6a shows the cross-sectional STEM image of WSe₂/phosphorene interface. Figure. R6b shows the elemental mapping for carbon (C), phosphorus (P), selenium (Se), and tungsten (W) of the cross section. The STEM image and elemental mapping demonstrate a clean and amorphous-phase-free WSe₂/phosphorene interface.

Description of STEM characterizations have been included in the main text (see Page 6). Figure R6 and detailed discussion have been included in Supplementary Information (see Supplementary Fig. 2 and Supplementary Note 2).

Figure R6 a, Cross-sectional scanning transmission electron microscopy (STEM) image of WSe₂/phosphorene interface. The scale bar is 2 nm. **b**, Elemental mapping of WSe₂/phosphorene interface.

3. *The authors assume that there is nearly 100% coupling at the hetero-interfaces. Was that confirmed? Can that be confirmed? How would a less than ideal coupling affects the results?*

Response: This is an insightful observation! We agree with you that the assumption of 100% coupling at the hetero-interfaces is not appropriate. In the revised manuscript, we have improved our model through taking account into the interlayer coupling coefficient at WSe₂/phosphorene (c_h) and phosphorene/phosphorene (c_p) interfaces, and the thermal expansion coefficient (TEC) and Young's modulus of WSe₂ and phosphorene layers. The coupling efficiency at WSe₂/phosphorene hetero-interfaces is not simply regarded as 100 % but depends on the model fitting results. Detailed simulation process is shown in Supplementary Note 3 and 4.

Figure R7 shows the calculated shear thermal deformation (STD) and interlayer shear thermal deformation (ISTD) in WSe₂/phosphorene heterostructure using previous (Figure R7a, b) and new (Figure R7c, d) models, respectively. Compared with previous results, the new STD of phosphorene layer starts to bend upwards near the WSe₂/phosphorene hetero-interface (Figure R7c) and ISTD does not vanish near the hetero-interface (Figure R7d). This phenomenon directly reflects the interlayer interactions between WSe₂ and phosphorene. The non-zero ISTD between WSe₂ and phosphorene indicates that the coupling at the hetero-interface is less than 100%. In the revised manuscript, Figure R7 is included in the main text (see Figure 5).

Figure R7. Layer-dependent shear thermal deformation (τ) and interlayer shear thermal deformation ($\Delta\tau$) of atomically thin layer in WSe_2 /phosphorene heterostructure using previous (a, b) and new theoretical models (c, d). Here, the hollow dots denote values in phosphorene while the solid dots denote values in WSe_2 .

In addition, the extracted thermal parameters of phosphorene from the new model are also updated in the revised manuscript (see Page 12-13 and Figure 6 in the revised manuscript). **Figure R8** compares the temperature-dependent TEC extracted from previous and new models. The $\text{TEC} \sim 45.2 \times 10^{-6} \text{ K}^{-1}$ at 300 K still agrees well with previously reported value between $6.3 \times 10^{-6} \text{ K}^{-1}$ and $53 \times 10^{-6} \text{ K}^{-1}$.

Figure R8. Temperature-dependent thermal expansion coefficient (TEC) of

phosphorene using previous and new models, respectively.

Point-by-point response to referee 3

We would like to thank you for your thoughtful comments and suggestions. We truly appreciate the time and efforts you invested in our paper. With your help, our paper has improved substantially. In this letter, we will respond to each of your comments following the order in which they appear in your report.

The manuscript by Le Zhang et al. discusses the interlayer shear strain in few-layer vdW materials. The authors developed an interesting idea to measure the strain of a few-layer 2D material by placing a monolayer WSe₂ on top and using its strain sensitive optical properties. The utilize their method to measure the thermal expansion of few-layer black phosphorous and hBN.

I see a couple of issues with the manuscript which need to be addressed:

1. In Fig. 1a the authors claim that the relaxation of the stress induced by the different thermal expansion coefficient of substrate and 2D material (which they call shear thermal deformation) from layer to layer is a novel thermomechanical phenomenon. While the measurement of the relaxation from layer to layer is new, the general picture is trivial. The reason that this effect has been neglected (not ignored) is, that it is rarely of high relevance in 2D heterostructures (e.g. in typical encapsulated hBN/1L TMD/hBN samples) or not measurable.

Response: Thank you for your valuable suggestions. The magnitude of thermal deformation does vary under different systems. We agree with you that in some cases, such as encapsulated hBN/1L-TMD/hBN systems, the thermomechanical deformation effect can be neglected, due to the small thermal expansion coefficient of hBN. Through carefully considering suggestions from all referees, we have improved the paper substantially, which not only shows an interesting optical approach for the measurement of layer-dependent relaxation, but also provides important interlayer coupling information at vdW homo- and hetero- interfaces. The improved theoretical model gives us a clear physical picture which not only can be used to explain the thermomechanical deformation, but also can be applied to various situations under mechanical bending or stretching. We summarized the major achievements of the revised version below:

First, thanks to the valuable suggestions from all referees, we have improved our theoretical model substantially through taking account into the interlayer coupling coefficient at WSe₂/phosphorene (c_h) and phosphorene/phosphorene (c_p) interfaces, and the thermal expansion coefficient (TEC) and Young's modulus of WSe₂ and phosphorene layers (see **Figure R9**). The new model not only can **quantitatively** resolve the STD and ISTD in individual layers of vdW materials (see **Figure R10**), but also is able to provide important information of interlayer interactions, such as the interlayer coupling coefficient (c_h , c_p) and in-plane force in each individual layer (**Figure R11**).

$\tau(n)$: shear thermal deformation (STD) of the n th layer
 $\Delta\tau(n)$: interlayer shear thermal deformation (ISTD)

Figure R9. The previous (left panel) and new (right panel) theoretical models. The new model considers the interlayer interactions between WSe₂ and phosphorene layers.

Figure R10. Layer-dependent shear thermal deformation (τ) and interlayer shear thermal deformation ($\Delta\tau$) of atomically thin layer in WSe_2 /phosphorene heterostructure using previous (a, b) and new theoretical models (c, d). Here, the hollow dots denote values in phosphorene while the solid dots denote values in WSe_2 .

Figure R11. Layer-dependent in-plane force in phosphorene (hollow circle) and WSe_2 (solid circle) layers extracted from the new model.

Second, thermal-induced strain is analogous to mechanical-induced deformation, but is more stable, reversible and controllable, which allows us to provide a clear physical picture and accurate results. Besides, our model can be applied to various mechanical deformation situations with various boundary conditions. Hence, our theoretical model and experimental results will provide useful information for researchers in 2D

material community.

Third, we develop a smart experimental strategy, using WSe₂-based heterostructures, to study the mechanical properties, thermal properties and interlayer properties of vdW materials.

Hence, we believe the new experimental methodology, the new model, the clear physical picture and the interlayer coupling information provided in this work will be appealing to broad researchers in 2D material community. In the revised manuscript, we have made substantial changes to the model, data fitting, and discussions, which are all highlighted in the main text. Figure R9 and R10 have been included in the main text (see Figure 1 and Figure 5) and Figure R11 has been included in the Supplementary Information (see Supplementary Figure 5).

2. While the authors claim that the strain is not fully transferred from layer to layer, they controversially assume "100 % strain transfer efficiency at the WSe₂/BP heterointerface" and "Thirdly, the attaching of WSe₂ has ignorable influence on the thermal deformation of phosphorene, in which case WSe₂ merely plays the role of strain sensor.", which definitely not true. Here, the authors need to discuss the Young's modulus of the BP and WSe₂ (which are pretty different) as well as the vdW coupling. In fact, on thin (1L, 2L,...) BP samples, the "probe" WSe₂ should have a significant impact on the expansion of the BP, so the TEC of the WSe₂ needs to be considered as well.

Response: These are insightful observations! Your comments really made us rethink the validity of these assumptions in our previous model. We totally agree with you that the interlayer coupling between WSe₂ and phosphorene and TEC of both phosphorene and WSe₂ must be considered when analyzing the shear thermal deformation (STD) and interlayer shear thermal deformation (ISTD).

In the revised manuscript, we have improved the theoretical model through taking account into the interlayer coupling coefficient at WSe₂/phosphorene (c_h) and phosphorene/phosphorene (c_p) interfaces, and the Young's modulus and thermal expansion coefficient (TEC) of WSe₂ and phosphorene (see **Figure R9** above). The boundary condition at WSe₂/phosphorene interface has been rewritten. The strain transfer efficiency is no longer treated as 100 %, but depends on the coupling coefficient c_h and TEC of phosphorene and WSe₂. Besides, we use accurate discrete method instead of previous continuous approximation method to conduct the calculations. The detailed simulation process is shown in Supplementary Note 3 and 4.

Here, we compare the simulation results of STD and ISTD using previous and new ISTD models at several representative layer numbers $N = 5, 10, 20, 30, 40$ and 50 as shown in **Figure R10** above. Comparing Figure R10a,b and Figure R10c,d, the new

model clearly reveals the non-uniform deformation of phosphorene layers near the WSe₂/phosphorene interface, which is the consequence of the interlayer coupling effect between WSe₂ and phosphorene layer and the large Young's modulus of WSe₂ (~120 GPa). As a result, there appears a crossover point where the negative $\Delta\tau$ turns into positive as shown in Figure R10d.

In the revised manuscript, we have included Figure R9 and Figure R10 into the main text (see Figure 1 and Figure 5). Corresponding discussions about the model and results are shown in Page 8-13 in the main text.

3. Furthermore, it is obvious, that the comparison between WSe₂ on SiO₂ and WSe₂ on BP fails already due to the different dielectric environments. Therefore, "Taking WSe₂ on SiO₂ as a reference system," is not a valid argument. However, the relative measurements on the differently thick BP should be ok.

Response: Thank you for pointing out this important issue to us. We do agree with reviewer that WSe₂ in WSe₂/SiO₂ and WSe₂/BP/SiO₂ experiences different dielectric environments, since the static dielectric constant of SiO₂ (~3.9)² is smaller than that of BP (~6)³. It is well known that the exciton binding energy of low-dimensional materials is sensitive to the dielectric environment. However, in this work, it is the $\Delta E' = \Delta E_{10K} - \Delta E_{300K} = (E_{WSe_2/BP}(10K) - E_{WSe_2}(10K)) - (E_{WSe_2/BP}(300K) - E_{WSe_2}(300K))$ that is used to determine the thermal deformation of BP and WSe₂ since $\tau \propto \Delta E'$. Here, $E_{WSe_2/BP}(300K)$ and $E_{WSe_2}(300K)$ are PL photon energies of WSe₂/BP/SiO₂ and WSe₂/SiO₂ at 300 K, respectively. $E_{WSe_2/BP}(10K)$ and $E_{WSe_2}(10K)$ are PL photon energies of WSe₂/BP/SiO₂ and WSe₂/SiO₂ at 10 K, respectively. As we can see from the above equation, the effect of exciton binding energy (which strongly depends on the dielectric environment) is probably canceled when calculating $\Delta E'$. Besides, we further experimentally demonstrate that the dielectric environment has negligible impact on the determination of $\Delta E'$.

In this experiment, we fabricated two **BP/WSe₂/SiO₂** (WSe₂ is sandwiched by BP and SiO₂) heterostructure samples (see **Figure R12a**), where the monolayer WSe₂ in WSe₂/SiO₂ and BP/WSe₂/SiO₂ experiences different dielectric environments (see Figure R12b). In addition, the in-plane lattice deformation of WSe₂ in both WSe₂/SiO₂ and BP/WSe₂/SiO₂ heterostructures can be regarded as zero due to the strong clamping effect of SiO₂ substrates. We monitored the temperature-dependent photon energy of BP/WSe₂/SiO₂ using that of WSe₂/SiO₂ as a reference. Here, ΔE is the relative shift of photon energy in BP/WSe₂/SiO₂ (or WSe₂/BP/SiO₂) compared with that in WSe₂/SiO₂, while $\Delta E'$ is the difference of ΔE between 300 and 10 K ($\Delta E' = \Delta E_{10K} - \Delta E_{300K}$). The temperature-dependent ΔE of the two BP/WSe₂/SiO₂ samples (the orange and olive scatters) and a WSe₂/BP/SiO₂ sample (the violet scatters) are displayed in Figure R11c. $\Delta E'$ in BP/WSe₂/SiO₂ heterostructures are extracted to be 0 and -3 meV, which are negligible compared to the large value ~ 47 meV in WSe₂/BP/SiO₂ heterostructures. This experiment proves that different

dielectric environments in $\text{WSe}_2/\text{SiO}_2$ and $\text{BP}/\text{WSe}_2/\text{SiO}_2$ heterostructures play a minor role in determining $\Delta E'$ and τ . Therefore, we think the argument "Taking WSe_2 on SiO_2 as a reference system," is still valid.

In the revised manuscript, we have discussed the effect of dielectric environment on determination $\Delta E'$ and τ (see Page 6 in the main text and Supplementary Note 1). We have also included Figure R12 in to the Supplementary Information (see Supplementary Figure 1).

Figure R12. **a**, Optical image of a $\text{BP}/\text{WSe}_2/\text{SiO}_2$ heterostructure. The scale bar is 10 μm . **b**, Schematic diagrams of $\text{WSe}_2/\text{SiO}_2$, $\text{BP}/\text{WSe}_2/\text{SiO}_2$ and $\text{WSe}_2/\text{BP}/\text{SiO}_2$ at 10 K. **c**, Temperature-dependent photon energy difference (ΔE) between WSe_2 in heterostructures and isolated WSe_2 on SiO_2 . Here, the orange and olive scatters denote $\text{BP}/\text{WSe}_2/\text{SiO}_2$ systems and the violet scatters denote $\text{WSe}_2/\text{BP}/\text{SiO}_2$ system. The $\Delta E'$ is 0 and -3 meV in the two $\text{BP}/\text{WSe}_2/\text{SiO}_2$ systems which can be neglected compared with the value ~ 47 meV in $\text{WSe}_2/\text{BP}/\text{SiO}_2$ system.

4. A value of the strain gauge factor $\eta = 100 \text{ meV}/\%$ for WSe_2 is taken from the literature. It needs to be discussed if this value is true for the whole temperature range from 10 to 300 K.

Response: This is a great point! We have performed experiments to investigate the

temperature dependence of the strain gauge factor for WSe₂. In order to measure the strain gauge factor, we exfoliate monolayer WSe₂ directly onto the polyimide (PI) membrane covered by a 50 nm-thick gold film (**Figure R13a**). We choose PI instead of PET membrane as the flexible substrate because PI can function well at low temperatures. Here, the 50 nm-thick gold film is sputtered onto PI membrane to eliminate the strong PL background from PI. Then, the WSe₂/Au/PI sample is loaded on a home-made strain setup with a two-point bending geometry, as shown in Figure R13b. Through pushing the side screw, the slider will move forward and PI membrane will be bended. A uniaxial tensile strain (ϵ) is therefore transferred to the monolayer WSe₂, which depends on the thickness ($t = 200 \mu\text{m}$) of substrate and the radius of curvature (R), $\epsilon = t/2R$.

Figure R13. **a**, Optical image of a monolayer WSe₂ on Au/PI membrane. The scale bar is 10 μm . **b**, Optical image of the experimental setup. **c**, Experimental measured (scatters) and theoretically fitted (dotted line) phonon energy of WSe₂ as a function of uniaxial tensile strain at 200, 250 and 300 K.

The strain setup is loaded in a He-flow closed-cycle cryostat with a high vacuum of $\sim 2 \times 10^{-6}$ Torr to conduct PL measurements. Noting that a severe PL quenching occurs when the monolayer WSe₂ is transferred onto Au layer. The PL intensity of monolayer WSe₂ decreases rapidly when temperature decreases. As a result, the PL signal of neutral exciton can only be distinguished above 200 K in our experiments. Hence, we can only provide the data above 200 K and we sincerely apologize for this. We plot the photon energy of WSe₂ as a function of tensile strain (0.23%, 0.45%, 0.61%, 0.77%) at 200, 250 and 300 K, respectively (scatters in Figure R13c). The fitting results show clear linear dependence between photon energy and strain (dotted line in Figure R13c). The biaxial strain gauge factor can be extracted to be -40, -46 and -42 meV/% at 200, 250 and 300 K, respectively. Hence, this experiment demonstrates that the strain gauge factor of WSe₂ shows weak temperature dependence, at least, in the range from 200 to 300 K.

	Ref. ⁴	Ref. ⁵	Ref. ⁶	Ref. ⁷	This work
Uniaxial (meV/%)	-54 (experiment)	-	-	-54 (experiment)	-21 (experiment)
Biaxial (meV/%)	-108 (estimated)	-63 (experiment)	-105 (estimated)	-108 (estimated)	-42 (estimated)

		-134 (theory)			
--	--	------------------	--	--	--

Table R3. Strain gauge factor of WSe₂ in previous reports and in this work.

Here, we list the strain gauge factor of WSe₂ obtained from our experiment and previous reports at room temperature in **Table R3**. It is clear that our experimental results are smaller than previously reported values, which could be attributed to the inefficient strain transfer at the WSe₂/Au interface. The real strain gauge factor is always underestimated in experiments. Therefore, in the manuscript, the biaxial strain gauge factor is adopted as -100 meV/%, which stands between the experimental and theoretical values. To further study the influence of strain gauge factor on the fitting results, the strain gauge factor is set to be -80, -100 and -120 meV/%. As shown in **Table R4**, both the interlayer coupling coefficients at WSe₂/phosphorene (c_h) and phosphorene/phosphorene (c_p) interfaces increase with $|\eta|$. Besides, c_h shows stronger η dependence than c_p . On the contrary, the intrinsic thermal deformation of phosphorene $|\tau_p|$ and TEC decrease with $|\eta|$.

η (meV/%)	c_h (Pa)	c_p (Pa)	τ_p	TEC at 300 K (K ⁻¹)
-80	1.94×10^{11}	3.22×10^{11}	-0.999 %	6.38×10^{-5}
-100	2.72×10^{11}	3.41×10^{11}	-0.705 %	4.52×10^{-5}
-120	3.50×10^{11}	3.63×10^{11}	-0.535 %	3.45×10^{-5}

Table R4. Fitting results when the biaxial strain gauge factor of WSe₂ is set to be -80, -100 and -120 meV/%, respectively.

In the revised manuscript, we have included the discussion of η in Supplementary Note 5. Table R3, Table R4 and Figure R13 have also been included (see Supplementary Table 1, Supplementary Table 2 and Supplementary Figure 3).

5. A minor point: The definition: "Firstly, the strain of the bottom phosphorene is negligible, considering the strong clamping effect between phosphorene and SiO₂ substrates¹⁸ and tiny TEC of SiO₂ substrate³³." is a bit counter intuitive from my point of view. I would rather argue that the SiO₂ transfers a strong (compressive) stress/strain to BP at low temperatures while the top layer of a high enough bulk BP is fully relaxed (i.e. no strain), especially, since the sample is prepared at room temperature, which means that the sample should be fully relaxed at 300 K (in contrast to the schematic drawing in Fig. 3c).

Response: Thank you for pointing out these issues to us. We agree with you that SiO₂ transfers a strong stress to the bottom phosphorene at low temperature. It is not appropriate to claim that: "the strain of the bottom phosphorene is negligible". Considering the TEC of phosphorene is at least one order of magnitude larger than that of SiO₂, the shear thermal deformation (STD) of SiO₂ can be regarded as zero compared with that of phosphorene from 300 to 10 K. Due to the strong clamping effect, STD of the bottom phosphorene is also negligible. To avoid misunderstanding,

in the revised manuscript, we claim that “the STD of the bottom phosphorene is negligible” instead of “the strain of the bottom phosphorene is negligible” (see Page 8-9 in the main text). We have also removed the schematic drawing in Figure 3c.

6. Everything said above is also true for the hBN measurement.

Response: In the revised manuscript, we have applied the new theoretical model to the WSe₂/hBN system and re-calculated all the parameters. The fitting results are shown in **Figure R14a**. The interlayer coupling coefficient at hBN/hBN homo-interface is extracted as $c_{\text{hBN}} = 5.03 \times 10^{11}$ Pa and intrinsic thermal deformation of hBN from 300 to 10 K is $\tau_{\text{hBN}} = 0.17\%$. The extracted thermal expansion coefficient (TEC) as a function of temperature is shown in Figure 14b. Compared with phosphorene, STD of hBN shows the opposite trend with total layer number N , which is attributed to the negative TEC value in hBN. In the revised manuscript, we have included Figure R14 and related discussion in the main text (see Figure 4b, Figure 6b, and Page 13).

Figure R14. **a**, Experimental measured and theoretically fitted STD (τ_h) of WSe₂ as a function of total layer number (N). **b**, Temperature-dependent thermal expansion coefficient (TEC) of phosphorene and hBN.

In conclusion: The measurement of the strain transfer (or strain relaxation) from layer to layer is interesting and can have implications for other groups working on strain engineering of 2D heterostructures. In principle the measurements should also give access to the interlayer vdW coupling / stacking order energies.

However, I do not see a broader interest as the fundamental effect is rather trivial. Furthermore, I see several critical issues with the manuscript that require a major revision.

Response: Thank you for your valuable constructive suggestions and for pushing us to think harder about the contribution of the work to 2D material community. We carefully considered all your suggestions and have improved the theoretical model substantially, which allows us to access the interlayer vdW coupling information in

2D heterostructures including interlayer coupling coefficient $c_p \sim 3.41 \times 10^{11}$ Pa, $c_h \sim 2.72 \times 10^{11}$ Pa and the layer-dependent in-plane stress (see Figure R11 above). Here, the in-plane stress in the n -th layer is caused through the interlayer shear interaction with adjacent layers. The schematic diagram in Figure R15 shows that the in-plane stress (σ) acts on the vertical section ($d \times W$) while interlayer shear stress (f) acts on the horizontal section ($l \times W$). According to Figure R15, the interlayer force per unit width ($F^* = F / W$) in the n -th layer can be described as $\sigma \times d$ and $f \times l$ equivalently. As shown in **Table R5**, the values of F^* extracted from our work is at the same order of previously reported values in graphene and carbon nanotube, demonstrating that our model is reliable and is able to extract interlayer coupling information from thermal deformation measurements in 2D heterostructures.

Figure R15. Schematic diagram of the in-plane stress in the n -th layer phosphorene. Here, the blue arrow indicates in-plane stress (σ) in our work while the orange indicates interlayer shear stress (f). d is the thickness of phosphorene (~ 0.55 nm) and l is the average characteristic length (~ 1 μ m).

	Ref. ⁸ graphene	Ref. ⁹ carbon nanotube	This work (at $n=50$)
F^* (N/m)	0.04	0.05	0.11

Table R5. Force per unit length ($F^* = F / W$) extracted from previous reports and this work.

Here, we briefly summarize the major contributions of this work. **First**, we provide a new experimental strategy, WSe₂-based heterostructure, to investigate rich physics of vdW materials, including mechanical and thermal properties. **Second**, the valuable suggestions from referees allow us to build an effective theoretical model to quantitatively describe the layer-dependent deformation of vdW materials and to access interlayer coupling information. **Third**, the model can be applied and extended to various deformation situations (both thermal-induced and mechanical-induced deformation) with various boundary conditions, which can be easily modified and used by other researchers. Hence, we believe the revised manuscript has improved substantially and will be beneficial to 2D material community.

Concluding remarks:

We would like to thank the entire editorial team once again for your very insightful comments and valuable suggestions, which helped us greatly improve the paper. We hope that we have addressed all of your comments to your satisfaction.

REFERENCES

1. Gong, L. et al. Reversible loss of Bernal stacking during the deformation of few-Layer graphene in nanocomposites. *ACS Nano* **7**, 7287-7294 (2013).
2. Robertson, J. High dielectric constant oxides. *Eur. Phys. J.: Appl. Phys.* **28**, 265-291 (2004).
3. Qiu, D. Y., da Jornada, F. H. & Louie, S. G. Environmental screening effects in 2D materials: renormalization of the bandgap, electronic structure, and optical spectra of few-Layer black phosphorus. *Nano Lett.* **17**, 4706-4712 (2017).
4. Schmidt, R. et al. Reversible uniaxial strain tuning in atomically thin WSe₂. *2D Mater.* **3**, 021011 (2016).
5. Frisenda, R. et al. Biaxial strain tuning of the optical properties of single-layer transition metal dichalcogenides. *npj 2D Mater. Appl.* **1**, 10 (2017).
6. Ahn, G. H. et al. Strain-engineered growth of two-dimensional materials. *Nat. Commun.* **8**, 608 (2017).
7. Cho, C. et al. Highly strain-tunable interlayer excitons in MoS₂/WSe₂ heterobilayers. *Nano Lett.* **21**, 3956-3964 (2021).
8. Wang, G. et al. Measuring interlayer shear stress in bilayer graphene. *Phys. Rev. Lett.* **119**, 036101 (2017).
9. Kis, A., Jensen, K., Aloni, S., Mickelson, W. & Zettl, A. Interlayer forces and ultralow sliding friction in multiwalled carbon nanotubes. *Phys. Rev. Lett.* **97**, 025501 (2006).

REVIEWERS' COMMENTS

Reviewer #1 (Remarks to the Author):

The authors have addressed my concerns with the paper and have made significant changes. The main issue I had was that their analysis did not take into account the stiffness of the WSe₂ layer. This is a point also raised by Reviewer #3. They have now modified their analysis to take this into account. They have also made a number of other changes and I feel that the paper is now suitable for publication.

Reviewer #2 (Remarks to the Author):

All my comments, questions and concerns have been carefully addressed by the authors. They have provided a new model, additional experimental data and I have no further objections to the publication of this manuscript as is.

Reviewer #3 (Remarks to the Author):

The revised version of the manuscript by Le Zhang et al. entitled "Probing interlayer shear thermal deformation in atomically-thin van der Waals layered materials" made substantial changes in modeling the shear thermal deformation in a phosphorene/WSe₂ (and hBN/WSe₂) heterostructures and added further information in the manuscript and SI. The new model seems to be correct, although it needs some external input parameters (biaxial strain gauge factor and Young's modulus of monolayer WSe₂), which are still not very well known in literature. Notably, the model now gives access to the interlayer coupling coefficients. All in all, I have to say, I appreciate the efforts taken by the authors to correct and improve the manuscript. In my opinion, the manuscript can be published in the current form and will add interesting information and a new method to the 2D materials community.